# Damage Detection in Largely Unobserved Structures under Varying Environmental Conditions: An AutoRegressive Spectrum and Multi-Level Machine Learning Methodology

**DOI:** 10.3390/s22041400

**Published:** 2022-02-11

**Authors:** Alireza Entezami, Stefano Mariani, Hashem Shariatmadar

**Affiliations:** 1Department of Civil and Environmental Engineering, Politecnico di Milano, Piazza L. da Vinci 32, 20133 Milano, Italy; stefano.mariani@polimi.it; 2Department of Civil Engineering, Faculty of Engineering, Ferdowsi University of Mashhad, Mashhad 9177948944, Iran; shariatmadar@um.ac.ir

**Keywords:** structural health monitoring, limited sensors, environmental variability, spectral estimation, Markov Chain Monte Carlo, factor analysis

## Abstract

Vibration-based damage detection in civil structures using data-driven methods requires sufficient vibration responses acquired with a sensor network. Due to technical and economic reasons, it is not always possible to deploy a large number of sensors. This limitation may lead to partial information being handled for damage detection purposes, under environmental variability. To address this challenge, this article proposes an innovative multi-level machine learning method by employing the autoregressive spectrum as the main damage-sensitive feature. The proposed method consists of three levels: (i) distance calculation by the log-spectral distance, to increase damage detectability and generate distance-based training and test samples; (ii) feature normalization by an improved factor analysis, to remove environmental variations; and (iii) decision-making for damage localization by means of the Jensen–Shannon divergence. The major contributions of this research are represented by the development of the aforementioned multi-level machine learning method, and by the proposal of the new factor analysis for feature normalization. Limited vibration datasets relevant to a truss structure and consisting of acceleration time histories induced by shaker excitation in a passive system, have been used to validate the proposed method and to compare it with alternate, state-of-the-art strategies.

## 1. Introduction

Civil structures must be monitored to detect, ideally in real time, any damage due to aging, material deterioration or unexpectedly large excitations. Structural health monitoring (SHM) systems provide means to assess the health and safety of civil, mechanical, and aerospace structures by exploiting various data such as vibration responses (e.g., acceleration time histories, modal data, strain, etc.) [1,2,3,4,5], images [6,7], and videos [8,9]. The primary step of SHM is the evaluation of the state of the monitored structure for damage detection purposes: this is known as early damage detection. The main goal of the aforementioned step is to seek whether damage has been triggered anywhere in the structural system. Although the implementation of early damage detection methods appears simple, the accuracy of the subsequent SHM steps (namely, damage localization and quantification) largely depends on the effectiveness of early damage detection.

Due to recent advances in sensing and data acquisition systems, the strategies in the SHM realm have been shifted from model-driven techniques under the concept of finite element model updating [10,11,12,13] to data-driven or data-based methods based on statistical pattern recognition and machine learning [1,14,15,16,17]. In contrast to model-based techniques that require elaborate numerical models of real-life structures, data-driven methods are only based on raw measurements with no requirement for numerical modeling and model updating strategies. In other words, the main objective of data-driven methods is to discover meaningful information (features) in the measured data and then use such features for decision-making within the context of machine learning [18]. Accordingly, these approaches basically consist of feature extraction and statistical analysis levels. Feature extraction focuses on delving into the measured vibration data to obtain certain damage-sensitive features. A damage-sensitive feature is any information extracted from the raw measurements, which must be sensitive to damage and not dependent on other factors, such as operational and environmental conditions. Since most of the data-based methods handle vibration signals, advanced signal processing techniques are necessary to extract damage-sensitive features from them [19].

The subsequent statistical analysis handles the obtained damage-sensitive features to make a decision concerning damage occurrence via statistical approaches. For this purpose, the feature datasets relevant to two different structural states must be compared. The process of damage detection via statistical analysis is thus based on the comparison between two structural states at different times, in order to identify discrepancies indicative of damage occurrence. To this aim, the most relevant techniques are statistical distance measures, which may depend upon the type of damage-sensitive features to handle. Some of the useful univariate and multivariate distance techniques to mention include the Mahalanobis distance [20,21,22] and Kullback–Leibler divergence [15,23,24], dynamic time warping [25], and other damage indices based on relative errors [26,27], classical and robust multidimensional scaling algorithms [28,29], etc.

An initial step of the entire vibration-based SHM strategy is related to the design of the case-specific sensor network, so as to capture sufficient dynamic information on the structure [30,31,32]. The effectiveness of the SHM system relies on the sensitivity to damage of any feature extracted from the sensed structural responses. This is typically attained with pervasive or dense sensor networks, so that the structural behavior results can be largely observed. Since structural damage directly affects and changes inherent structural properties, particularly stiffness, a damage-sensitive feature is claimed to also be relevant to the structural properties or their variations. An important issue in SHM applications is thus represented by the preliminary design of the sensor deployment, to provide observations or measurements at specific locations [33]. Although advances in sensing technology can enable the implementation of a large number of sensors in the network, their cost and supporting instruments may represent serious obstacles. Furthermore, a majority of civil structures in need of SHM are complex and large-scale, and the installation of several sensors may not be trivial and affordable. Due to such circumstances, it is inevitable that the SHM procedure is carried out by exploiting information acquired by a limited number of sensors only [34,35,36,37].

Overall, despite the success and applicability of various feature extraction and statistical analysis techniques, the adoption of a limited number of sensors may prevent these approaches from capturing sufficient dynamic characteristics or damage-sensitive features to provide reliable damage detectability. This issue becomes even worse when the limited information is coupled with the environmental and/or operational variability conditions. These are deceptive effects, such as temperature fluctuations, humidity and moisture variations, wind speed, human movements, and traffic, that provide changes similar to damage in the sensed structural response and lead to an outlier masking problem [21]. In such cases, false alarms and erroneous detection results present the major challenges [20,38,39]. On the other hand, depending upon the type of damage sensitive feature used in SHM, the level of variability can fluctuate [2].

In order to deal with the aforementioned limitations and challenges, this article proposes a parametric spectral-based feature extraction approach and an innovative multi-level machine learning method for early damage detection in cases characterized by a limited number of sensors and under environmental variability. Hence, the main objective of this research is to assess whether a structure with limited information collected through the sensors, has actually been affected by damage or is still in its normal state. The proposed method of feature extraction is based on an autoregressive (AR) representation, to model the measured vibrations in the time-domain and estimate their spectra as damage-sensitive features by means of the Burg method. The proposed machine learning method consists of three main levels: (i) distance calculation by the log-spectral distance (LSD) to increase damage detectability and generate distance-based training and test samples, (ii) removal of environmental variability via feature normalization by an improved factor analysis with Markov Chain Monte Carlo (MCMC) and Hamiltonian Monte Carlo (HMC) sampler, or with a sampling called MCMC-FA to deal with the classical factor analysis, and (iii) decision-making for damage localization by a relative entropy measure called Jensen–Shannon (JS) divergence.

The improved factor analysis aims at dealing with the limitation of the covariance matrix estimation, which represents an important item in factor analysis, when the multivariate dataset is low-dimensional. In this case, the estimate of the covariance matrix may become problematic [40]. Therefore, the proposed method exploits the merits of the MCMC technique and of Hamiltonian sampling to increase the size of the multivariate data, and guarantees an appropriate estimate of the covariance matrix. The major contributions of this article are given by the development of an innovative multi-level machine learning method coping with the issue of limited sensor deployment and under environmental variability, and the proposal of an improved factor analysis for removing any variability in the data. The effectiveness and performance of the proposed method are assessed through limited vibration data relevant to a laboratory truss structure, known as the Wooden Bridge. Some comparative analyses were carried out to demonstrate the superiority of the methods presented in this article over existing techniques. Results have shown that the use of the AR spectrum provides a more reliable result in terms of damage detection, as compared to the direct use of AR coefficients in the case of limited information. Above all, it has been observed that the proposed multi-level machine learning method is able to accurately detect damage in cases characterized by a limited sensor deployment and environmental variability, owing essentially to the improved factor analysis approach.

## 2. Parametric Spectral-Based Feature Extraction by AR Modeling

Spectral analysis may provide several approaches to characterize the frequency content of a signal; all of them are based on estimating the power spectral density (PSD) of the signal from its time-domain representation. The output spectral density thus characterizes the frequency content of the said signal, managed as a stochastic process [41].

Spectral analysis can be carried out by means of either non-parametric or parametric methods. Non-parametric techniques, such as the FFT-based Welch’s method or periodogram, do not require prior detailed knowledge of the signal. The main advantage of these methods is the capability to handle any kind of signal. Parametric methods, such as Burg, covariance, and MUSIC ones, are model-based approaches that incorporate instead a prior knowledge of the signal and can therefore yield more accurate spectral estimates. A model to generate the signal can be based on a number of parameters that must be estimated from the observed data. Departing from the model and estimated parameters, any algorithm computes the relevant, model-dependent power density spectrum. These methods thus estimate the PSD by first tuning the parameters of the (linear) system able to generate the signal to handle. Parametric methods perform better than non-parametric ones, with an additional tendency towards higher resolutions.

The AR representation is commonly used for linear systems in parametric spectral-based approaches. Such a model of a stationary stochastic process is known as an all-pole one in signal processing sense, or a filter with all its zeroes at the origin of the *z*-plane. Given the sensed vibration signal *y*(*t*), the AR model reads:(1)$$y\left(t\right)+\theta_{1}y\left(t-1\right)+\cdots +\theta_{p}y\left(t-p\right)=r\left(t\right)$$
where *r*(*t*) is an independent, identically distributed stochastic sequence with zero mean at time *t*, known as the residual error. In Equation (1), *p* is the order of the AR model, and *θ*_1_…*θ_p_* are the model coefficients to be estimated to fit the observations. By exploiting the model order and coefficients, the AR spectrum can be estimated [42]. To set the model order, in this work the iterative methodology proposed in Entezami and Shariatmadar [26] was adopted: this methodology rests on a residual analysis via the Ljung–Box Q-test, and the AR order is chosen to satisfy the main criteria of the aforementioned hypothesis test.

One of the most effective approaches to estimate the AR spectrum is the Burg method. Compared to other AR spectral estimation techniques, the Burg method bears the main advantages of resolving closely spaced sinusoids in signals with low noise levels, and estimating short data records with high accuracy [43]. In addition, the method assures a stable AR model within a computationally efficient parameter estimation procedure. Overall, the method is based on the minimization of the forward and backward prediction errors, also satisfying the Levinson-Durbin recursion [44]. It avoids the calculation of the autocorrelation function by estimating the reflection coefficients directly. In concrete terms, the *p*th reflection coefficient is a measure of the correlation between *y*(*t*) and *y*(*t* − *p*), once the correlation due to the observations *y*(*t* − 1) … *y*(*t* − *p* + 1) has been filtered out; these reflection coefficients can be transformed into autoregressive parameters by means of the Levinson-Durbin recursion formula. Accordingly, here the AR spectrum *P*(*ω*) is estimated through:(2)$$P\left(\omega \right)=\frac{\sigma_{r}^{2}}{{\left|1-\sum_{k=1}^{p}\theta_{k}e^{-j\omega k}\right|}^{2}}$$
where $\sigma_{r}^{2}$ denotes the variance of the model residuals.

## 3. Proposed Multi-Level Machine Learning Method

The proposed machine learning method is composed of three main levels. The first level aims at calculating the distance between two spectra obtained for the training and test datasets, respectively, relevant to structural states in the baseline and monitoring phases, by using the LSD. In this regard, one can generate new damage-sensitive features from the initial features, that is the AR spectra. Since these new features originate from the distance calculation procedures relevant to the initial ones, the damage detectability in case of a limited number of sensors is expected to enhance.

To manage the negative effects of environmental variability in the second level, the proposed MCMC-FA method is employed to remove such variability in the distance-based features. Finally, the third procedure level exploits the normalized features provided by the MCMC-FA, to detect damage via the JS-divergence with the aid of the classical kernel density estimation (KDE). To clarify the entire procedure, Figure 1 depicts the flowchart for feature extraction and the three-level machine learning method. In the third level, once the distance values of the features regarding the normal conditions have been determined, an alarming threshold for decision-making is also estimated [38,45,46]. To this aim, the proposed strategy exploits the threshold estimation method proposed in Sarmadi and Yuen [47] on the basis of the extreme value theory, and the technique of peak-over-threshold.

### 3.1. Level I: Training and Test Data Generation by Log-Dpectral Distance

As previously mentioned, the main objective of the first level of the procedure is to provide multivariate training and test datasets, using the AR spectra related to the normal and current states of the structure. For this purpose, it is necessary to calculate the (dis)similarity between those spectra via the LSD, which is a symmetric distance measure able to compute the discrepancy between two sets of frequency-domain data [48]. Given the two spectra *P*(*ω*) and $\bar{P}$(*ω*), the LSD is given by:(3)$$\mathrm{LSD}=\sqrt{\frac{1}{2\pi }\int_{-\pi }^{\pi }{\left|\log\;\bar{P}\left(\omega \right)-\log P\left(\omega \right)\right|}^{2}d\omega }=\sqrt{\frac{1}{2\pi }\int_{-\pi }^{\pi }{\left|\log\frac{\bar{P}\left(\omega \right)}{P\left(\omega \right)}\right|}^{2}d\omega }$$

If the spectra are discrete, the LSD is provided as follows [49], p. 365:(4)$$\mathrm{LSD}=\sqrt{\frac{1}{n_{p}}\sum_{i=1}^{n_{p}}{\left|\log\;\bar{P}\left(i\right)-\log P\left(i\right)\right|}^{2}}=\sqrt{\frac{1}{n_{p}}\sum_{i=1}^{n_{p}}{\left|\log\frac{\;\bar{P}\left(i\right)}{P\left(i\right)}\right|}^{2}}$$
where *n_p_* denotes the number of spectrum samples. The LSD value equals zero if and only if the two spectra *P*(*ω*) and $\bar{P}$*(ω*) are exactly the same; therefore, any difference between them leads to an LSD value larger than zero. In case *P*(*ω*) and $\bar{P}$(*ω*) are the AR spectra at a specific sensor location, respectively, associated with the normal and current states of the structure, a deviation of $\bar{P}$(*ω*) from *P*(*ω*) is likely indicative of damage occurrence.

The above-mentioned procedure is based on the distance calculation between two spectra. In reality, there is more than a single sensor mounted on the structure to sense its response to external actions, and dynamic tests are repeated several times to collect data measurements. If *n_s_* sensors are deployed over the structure and if the dynamic test is repeated *n_m_* times, indices *S*_1_…*S_c_* are used denote the *n_c_* normal conditions of the structure in the baseline phase: therefore, the training dataset is given by **X** ∈ $R^{n_{x}\times n_{s}}$, where *n_x_* = *n_m_*×(*n_c_*−1). As the distance of any spectrum from itself in a normal condition is always zero and does not need to be accounted for in the analysis, in what precedes *n_c_*−1 data collections are considered to provide *n_x_*. It is worth remarking that each column of **X** is the LSD between the spectra corresponding to two different normal conditions at the same sensor location.

Now, let *S_u_* be the current structural condition, for which the health of the structure in terms of damage occurrence must be monitored. Distance calculation is carried out now by computing the LSD between the spectra corresponding to the normal and current states at the same sensor location. This procedure is repeated to obtain all distance values for the *n_c_* normal conditions at the *n_s_* sensor locations, for all the *n_m_* measurements. The test matrix is thus obtained as **Z**∈$R^{n_{z}\times n_{s}}$, where *n_z_* = *n_m_* × *n_c_*.

### 3.2. Level II: Feature Normalization by MCMC-FA

Due to the effects of environmental and/or operational variability in real-world circumstances, it is essential to remove such variability from the data so as to provide more reliable features sensitive to damage only. In the context of SHM, the removal of environmental and/or operational variability is often carried out by certain technical strategies called *data* or *feature normalization* [18]. To avoid any confusion regarding the removal of the aforementioned variability from measured data (i.e., acceleration time histories) or features (e.g., modal data or statistical characteristics of the time series), the term feature normalization is here adopted to refer to procedures aiming to eliminate environmental and/or operational variability from features extracted from measurements.

Such procedures are often implemented by handling a linear model as the residual or difference between the original features including the environmental and/or operational variability, and the output features of a machine learning model trained ad hoc. In this regard, the aforementioned final features contain all the information about structural damage, which must be distinguished from the variability of the context [18].

#### 3.2.1. Classical Factor Analysis

Factor analysis is a statistical technique to analyze multivariate data, which either aims to describe the variability among observed variables or to reduce the dimension of data. In both cases, the goal is attained in terms of a potentially low number of unobserved variables called *latent variables* or *factors*. The main objective of factor analysis is to develop a linear model to identify such unobservable variables [50].

Given the multivariate data **X**∈$R^{n_{x}\times n_{s}}$, the linear factor model can be expressed in matrix form as:(5)X=ΛΨ+E
where: **Λ**∈$R^{n_{x}\times n_{f}}$ is the matrix that includes factor loadings; **Ψ**∈$R^{n_{f}\times n_{s}}$ is the matrix of the latent variables or factor scores; and **E**∈$R^{n_{x}\times n_{s}}$ is the matrix of residuals or errors in factor modeling, which are assumed to be independent of the factor scores. Within the field of SHM, factor analysis is used to normalize features by removing the variability in the initial data. The covariance matrix **Σ** of the initial data or feature set **X** must be estimated, and decomposed in the following form:(6)$$\Sigma =\Lambda \Lambda^{T}+\Phi$$
where **Φ** is a diagonal matrix gathering the specific variances. To obtain **Λ** and **Φ**, the maximum likelihood estimation method can be adopted via the expectation maximization algorithm [50]. Once these matrices have been determined, the matrix of factor scores is obtained as follows:(7)$$\Psi =\Lambda^{T}{\left(\Phi +\Lambda \Lambda^{T}\right)}^{-1}X$$

Factor analysis or any other regression modeling approach has the ability to reconstruct the initial data or generate the independent residual data. Since the matrices of the factor loadings and of scores are known, the feature normalization model of the features of the normal condition can be defined as **E_x_** = **X − ΛΨ**. Through this expression, the variability in the initial data **X** caused by the environmental and/or operational conditions can be removed.

The same linear model can be used to eliminate the variability in the features regarding the current state of the structure in the monitoring phase. Given the feature matrix **Z^*^**∈$R^{n_{x}\times n_{s}}$, the same estimated matrices of the factor loadings and scores can be used to obtain the residual matrix **E_z_** = **Z^*^** − **ΛΨ**; note that **Z^*^** is a subset of **Z**, which contains *n_c_* sets of **Z**^*^. The residual matrices **E_x_** and **E_z_** can now be considered as the structural damage-sensitive features, to be adopted at the decision-making level.

#### 3.2.2. Markov Chain Monte Carlo Factor Analysis

The classical factor analysis is suitable for multivariate data when the dimension of sampling is high; in other words, this technique is applicable for dimensionality reduction, further than for data normalization. A main limitation of this approach may be the size of the multivariate data: in most cases, it is assumed that data of interest are high-dimensional, with a normality property of the factors [51]. It may then become problematic to estimate a covariance matrix when the multivariate data of interest are low-dimensional with a non-Gaussian distribution. The present work is intended to exploit the classical factor analysis for low-dimensional multivariate data, owing to the use of the MCMC and HMC samplers. The core of the proposed MCMC-FA method is to generate random Gaussian samples from the available data, and estimate the covariance matrix of such extended multivariate data.

In probability theory, MCMC is a computer-driven sampling method that allows the characterization of a probability distribution model by randomly sampling values out of the distribution of interest, without a thorough knowledge of its mathematical properties [51]. The term “Monte Carlo” refers to the practice of estimating the properties of a distribution by examining random samples obtained from the distribution of interest. The term “Markov Chain” refers instead to a sequential process of random sample generations, leading to new samples depending only on those immediately preceding the current ones [52].

The HMC sampler is a gradient-based MCMC method that draws samples from a target probability density, which is the multivariate Gaussian distribution here, of the low-dimensional data **X** [53]. The HMC sampler is based on a logarithmic function of the target distribution, its gradient and a momentum vector **λ**. Using these features, a Hamiltonian function *H*(**X**,**Ω**) based on Hamiltonian dynamics is defined as follows:(8)$$H\left(x,\lambda \right)=U\left(x\right)+V\left(\lambda \right)$$
where: **x**∈**X**; *U*(**x**) denotes the logarithmic function of the probability of interest; and *V*(**λ**) = ½ **λ**^T^**M**^−1^**λ, M** is a symmetric positive definite matrix that is typically diagonal or a scalar multiple of the identity matrix. *V*(**λ**) can be the opposite of the logarithmic probability density of the zero-mean Gaussian distribution with covariance matrix **M** [53]. Accordingly, Hamiltonian dynamics operates on **x** and **λ** to develop equations that aim at determining how the vectors **x** and **λ** change over time in the following forms:(9)$$\frac{dx_{k}}{dt}=\frac{\partial H}{\partial \lambda_{k}}=\frac{\partial V\left(\lambda \right)}{\partial \lambda_{k}}$$
(10)$$\frac{d\lambda_{k}}{dt}=-\frac{\partial H}{\partial x_{k}}=-\frac{\partial U\left(x\right)}{\partial x_{k}}$$
where *k* = 1,2,…,*n_s_*. By means of Equations (9) and (10), and relevant initial conditions in terms of **x**_0_ and **λ**_0_ at time *t*_0_, it is possible to simulate the evolution of the vectors via the leapfrog method [53].

The overall aim of the procedure is to predict **X** for pre-defined chain (*C*) and sampling (*N*) numbers. As such, the HMC algorithm is used to draw *N* samples of **X** from the target probability distribution, designated here as $\tilde{X}\in R^{N\times n_{s}}$, and *C* chains within an iterative strategy (i.e., *i* = 1,…,*C* and *j* = 0,…, *N*−1 for sampling ${\tilde{X}}_{j+1}^{\left(i\right)}$) and with the acceptance probability criterion from the Gelman–Rubin convergence statistic [54]. If convergence is attained, the simulated parameters at the (*j* + 1)th iteration is fixed; otherwise, the simulated parameters of the *j*th iteration should be selected. Once the *C* sets of $\tilde{X}$ have been determined, the average of these *C* sets is considered as the final multivariate datum for the covariance estimation. Using the estimated covariance matrix $\tilde{\Sigma }$, the procedure of Section 3.2.1, namely the decomposition of the covariance matrix $\tilde{\Sigma }$ into the matrices $\tilde{\Lambda }$ and $\tilde{\Phi }$ and the estimation of the matrices $\tilde{\Lambda }$ and $\tilde{\Psi }$, is adopted to set the new residual matrices $\tilde{E}$**_x_** = $-\tilde{\Lambda }\;\tilde{\Psi }$ and $\tilde{E}$**_z_** = $-\tilde{\Lambda }\;\tilde{\Psi }$.

The same HMC sampling procedure is next carried out for the multivariate data relevant to the current state, namely for **Z**, in order to set $\tilde{Z}\in R^{N_{z}\times n_{s}}$, where *N_z_* = *N*×*n_c_*. The extraction of the residual matrix $\tilde{E}$**_z_** is carried out for each of the *n_c_* sets of $\tilde{Z}$.

#### 3.2.3. Determination of the Number of Factors

Factor analysis is a parametric statistical approach, wherein the number of factors is the main unknown parameter to be determined. Although some analytical methods were proposed to tackle this issue [55,56], they are not related to the main challenges of this study, namely the limited number of sensors and the effects of environmental variability. For this reason, an effective approach is proposed to determine the number of factors of the MCMC-FA method with a specific focus on the aforementioned challenges.

First, as the vibration data collected with few sensors only are considered, the number of factors is fixed to also remain limited. Only the vibration data relevant to the normal conditions are assumed available; the number *n_f_* of factors must be thus compatible with the number *n_s_* of sensors, so that *n_f_* < *n_s_*. Second, the detrimental effects of the environmental variability on the process of decision-making must be assured to be minimized. In [47], it was demonstrated that false positive and false negative solutions are linked to the variability in the output of the decision-making level; therefore, *n_f_* is set to guarantee that the decision-making bears the minimum variance. An iterative procedure is thus proposed to determine *n_f_* by evaluating the variance of the output of the decision-making process, to finally keep the value corresponding to the minimum variance. For brevity, the entire procedure is illustrated in Figure 2 through the corresponding flowchart.

### 3.3. Level III: Decision-Making by Jensen-Shannon Divergence

Decision-making via statistical distance measures represents one of the most effective and efficient strategies due to its simplicity, computational efficiency, and non-parametric properties. Depending upon the types of data (features) to handle, being univariate vs. multivariate, random vs. deterministic, probabilistic vs. non-probabilistic, correlated vs. uncorrelated properties, there exist numerous measures that can be adopted for decision-making [49]. Having considered the multivariate datasets $\tilde{E}$**_x_** and $\tilde{E}$**_z_** provided by feature normalization, we propose to use the JS-divergence method with the aid of KDE for early damage detection. Although the KL-divergence is the most popular probabilistic measure, its non-symmetric and infinity properties are the main reasons to seek alternate solutions, such as e.g., the JS-divergence, which is instead a symmetric measure.

#### 3.3.1. Relative Entropy Measures in Information Theory

The KL and JS divergences fall into the family of statistical measures based on information theory and Shannon entropy. Shannon entropy is the main concept of information theory, and pertains to a single random variable or a random vector. More precisely, the entropy of a random variable can be defined as either a measure of the uncertainty of that variable, or a measure of the amount of information required on average to describe the random variable itself [57]. To develop this theory for decision-making, the KL-divergence method, called relative entropy measure, was proposed to compute the dissimilarity between two probability distributions, as it gives a measure of the extent to which the probability distribution of interest deviates from a reference one.

Given the probability distributions **p** = [*p*_1_,…, *p_n_*] and **q** = [*q*_1_,…, *q_n_*], where *n* > 1, the KL divergence is given as follows:(11)$$d_{\mathrm{KL}}\left(p,q\right)=\sum_{i=1}^{n}p_{i}\ln\left(\frac{p_{i}}{q_{i}}\right)$$
where *d_KL_*(**p**,**q**) > 0, being *d_KL_*(**p**,**q**) = 0 if and only if **p** = **q**. Due to Equation (11), it must be assumed that *q_i_* ≠ 0 for every *i*. One of the main drawbacks of the KL divergence is its non-symmetric behavior, which means that *d_KL_*(**p**,**q**) ≠ *d_KL_*(**q**,**p**); hence, this measure cannot satisfy all the conditions of a distance metric. Another drawback of the KL divergence is that it may diverge to infinity depending on the underlying probability distribution [58]. To avoid those limitations for decision-making, it is possible to adopt the JS-divergence. This entropy-based measure can be interpreted as the total KL-divergence away from the average distribution. For the probability distributions **p** and **q**, the JS-divergence is written as:(12)$$d_{JS}\left(p,q\right)=\frac{1}{2}\sum_{i=1}^{n}p_{i}\ln\left(\frac{2p_{i}}{p_{i}+q_{i}}\right)+q_{i}\ln\left(\frac{2q_{i}}{p_{i}+q_{i}}\right)$$
where *d_JS_*(**p**,**q**) > 0, *d_JS_*(**p**,**q**) = 0 if and only if **p** = **q**, and *d_JS_*(**p**,**q**) = *d_JS_*(**q**,**p**). Due to these properties and to the fact that the JS-divergence always displays a finite value, it can be adopted as a distance metric.

#### 3.3.2. Damage Detection Scheme

To detect damage via the JS-divergence, the probability distribution of each feature vector in **E_x_** and **E_z_** must be computed. By assuming that **p** and **q**, respectively, refer to the normal and current states, the main aim is now to link them with feature samples of **E_x_** and **E_z_**.

In the case of any prior knowledge about the distributions of the feature samples, the best method to obtain the aforementioned probability distributions is to use the KDE. This technique is based on a smoothing function and on a bandwidth value, which controls the smoothness of the resulting density curve. For the variable **x**, the probability density function determined by the kernel estimator is given by [59]:(13)$$f\left(x\right)=\frac{1}{nb}\sum_{i=1}^{n}K\left(\frac{x-x_{i}}{b}\right)$$
where: *x*_1_,…, *x_n_* are random samples obtained from an unknown distribution; *n* is the sample size; *K*(.) denotes the kernel smoothing function, which has to satisfy $\int_{-\infty }^{\infty }K\left(x\right)\mathrm{dx}$ = 1; and *b* > 0 is a smoothing parameter, called bandwidth. The most common kernel smoothing functions are the uniform, Gaussian, Epanechnikov, bi-weight, tri-weight, and triangle ones. The bandwidth of the kernel is a free parameter, with a significant effect on the final estimate; the most common optimality criterion used to select it is the mean integrated squared error [59].

After estimating the probability distributions of the feature vectors in $\tilde{E}$**_x_** and $\tilde{E}$**_z_**, the JS-divergence is adopted to determine the distance between the probability distributions. This procedure is implemented first for the baseline, and then for the monitoring phase. In the first phase, the distance between feature vectors in $\tilde{E}$**_x_** is computed to obtain the distance values relevant to the normal condition, which are next used to estimate the alarming threshold. In this work, the threshold estimation method proposed in Sarmadi and Yuen [47] was adopted, by exploiting the extreme value theory and the peak-over-threshold technique under the generalized Pareto distribution for threshold estimation. In this way, no false alarms in decision-making using the available (training) data are guaranteed to occur. Second, in the monitoring phase the distance between the probability distributions relevant to feature vectors in $\tilde{E}$_x_ and $\tilde{E}$**_z_** is performed. Accordingly, if the current state results to be damaged, the distance values are expected to exceed the estimated alarming threshold; if this does not occur, then the current state can be targeted as undamaged [38].

## 4. Case Study: The Wooden Bridge

The effectiveness and reliability of the proposed method are now assessed with a series of vibration measurements relevant to a laboratory truss structure under actual environmental variability. The structure is known as the Wooden Bridge [60], and is shown in Figure 3. The bridge was equipped with 15 accelerometers, whose deployment is also depicted in the figure. The sensors measured the acceleration time histories at three different longitudinal positions, and an electro-dynamic shaker was used to excite the structural vertical, transverse, and torsional modes under a random excitation source. The acceleration responses were all comprised of 8192 data points, evenly spaced during 32 s of data recording with a sampling frequency of 256 Hz. The measurements were collected over three days (18, 25 and 29 May), and represent undamaged and damaged cases under varying environmental conditions in terms of temperature and humidity [15]. All test measurements carried out on the first two days, and a few on the third day were representative of the normal condition of the bridge, see Table 1. All algorithms discussed in this study have been implemented in MATLAB R2017a.

The effects of damage in the bridge were fictitiously obtained by adding a mass at the end of the girder, close to Sensor 4. The added mass varied in the range 23.5–193.7 g, in order to represent a varying severity of damage, see Table 1. The number of test measurements *n_t_* was set to 20, for both the undamaged and the damaged states. According to common procedures of machine learning, the first two undamaged states of the structure (i.e., HC1, HC2 in Table 1) were considered in the baseline phase, so that *n_c_* = 2. The structural states HC3 and DC1-5 were instead exploited in the monitoring period, as current states. Accordingly, the goal of this analysis is to show whether the proposed method is able to detect HC3 and DC1-DC5 as the normal and damaged conditions, respectively.

### 4.1. Response Modeling and Feature Extraction

As detailed in Section 2, the first step of response modeling by the AR representation is the determination of the model order for each vibration signal, sensor location and test measurement. Only the vibration datasets relevant to states HC1 and HC2 in the training phase were handled to determine the said model orders. Next, the average order at each sensor location was adopted in the monitoring stage for states HC3, and DC1-DC5. By adopting the iterative approach proposed in [26], results are reported in Figure 4 in terms of the average AR order for Sensors 1–15 and for all test measurements. Using the obtained orders, the AR spectrum at each sensor location was estimated by means of the Burg method for all structural states. Figure 5 shows exemplary results related to the AR coefficients for Sensor 4, for the first test measurement of HC1, DC1 and DC5 states. It is worthy to note that, since the Wooden Bridge was excited by a shaker (namely, a measurable excitation condition), an AR representation was adopted to model the vibration responses. In case of ambient vibrations (namely, unmeasurable excitation conditions), time series models for response modeling should be based on error polynomial functions such as ARMA and ARARX [1,29].

The main purpose of this comparison is to assess the sensitivity to damage of these features, understanding that the states DC1 and DC5 are characterized by the smallest and largest damage severity, respectively. From Figure 5a, it remains difficult to identify any difference between the coefficients of the AR models regarding HC1 and DC1, while there is a clear difference between the two sets in Figure 5b. Although the comparison between the AR coefficients of HC1 and DC5 reveals the sensitivity to damage, the conclusion regarding Figure 5a emphasizes the necessity of applying a robust statistical approach for early damage detection.

### 4.2. Damage Detection with Limited Sensor Deployment and Under Environmental Effects

In order to detect damage in the Wooden Bridge in the case of a limited number of sensors deployed over the structure, four scenarios were defined and reported in Table 2. The first scenario is characterized by all sensors deployed over the structure handled for measurements, and here represents a reference situation of a partially observed structure equipped with a monitoring system (putatively) able to capture damage inception. The second and third scenarios are characterized by a decreasing number of sensors (roughly 50% of all deployed sensors); the difference between the two links to the fact that the third scenario does not consider the data obtained with Sensors 4 and 10, which are closest to the damaged area. Finally, the last scenario allows for only roughly 25% of the deployed sensors, with no data taken from sensors installed on the damaged area. For each scenario, the AR spectrum was built on the basis of the considered sensors only; hence, the goal of this investigation is to show how the proposed multi-level machine learning method can robustly provide an SHM strategy.

At the first level of the proposed method, the initial training datasets for the four scenarios are, respectively, represented by **X**_1_ ∈ ℝ^20 × 15^, **X**_2_ ∈ ℝ^20 × 7^, **X**_3_ ∈ ℝ^20 × 7^, and **X**_4_ ∈ ℝ^20 × 4^, where *n_x_* = 20 and *n_s_* = 15, 7, 7 and 4, due to the decreasing number of measurements allowed for. To obtain these matrices, the distance calculation was implemented to determine the LSD values between the AR spectra of states HC1 and HC2. Next, the distance calculation was implemented to measure the LSD values between the AR spectra related to each of the current states, and to the aforementioned normal conditions. The test datasets for the current states in the four scenarios are accordingly given by **Z**_1_∈ℝ^240×15^, **Z**_2_∈ℝ^240 × 7^, **Z**_3_∈ℝ^240 × 7^, and **Z**_4_∈ℝ^240 × 4^, where the number of rows (240) was obtained by the six sets of the test matrices, each consisting of 40 entries due to *n_z_* = *n_m_* × *n_c_* = 20 × 2.

The proposed MCMC-FA method was then adopted to remove any potential environmental variability from the distance-based features extracted at the first level of the strategy. The aim was also to extend the training and test matrices by the MCMC and HMC sampler, to better estimate the covariance matrices of **X**_1_–**X**_4_. The main features of the sampling process are listed in Table 3. Note that the implementation of the sampling process was based on the default functions hmcSampler, tuneSampler and drawSamples of MATLAB R2017a.

After the sampling process, 10 sets of the new extended training matrices were obtained. By averaging these sets, the extended training matrices ${\tilde{X}}_{1}\in R$^1000 × 15^, ${\tilde{X}}_{2}\in R$^1000 × 7^, ${\tilde{X}}_{3}\in R$^1000 × 7^, and ${\tilde{X}}_{4}\in R$^1000 × 4^ were arrived at. The same sampling procedure was implemented to generate the new extended test matrices ${\tilde{Z}}_{1}\in R$^2000 × 15^, ${\tilde{Z}}_{2}\in R$^2000 × 7^, ${\tilde{Z}}_{3}\in R$^2000 × 7^, and ${\tilde{Z}}_{4}\in R$^2000 × 4^ for each current state in the monitoring phase, where now *N_z_* = *N* × *n_c_* = 1000 × 2. Using these extended training matrices, the four covariance matrices ${\tilde{\Sigma }}_{1}\in R$^15×15^, ${\tilde{\Sigma }}_{2}\in R$^7 × 7^, ${\tilde{\Sigma }}_{3}\in R$^7 × 7^, and ${\tilde{\Sigma }}_{4}\in R$^4 × 4^ were set. The factor loading and score matrices $\tilde{\Lambda }$ and $\tilde{\Psi }$, and the residual matrices $\tilde{E}$**_x_** = −$\tilde{\Lambda }\;\tilde{\Psi }$ and $\tilde{E}$**_z_** = $-\tilde{\Lambda }\;\tilde{\Psi }$ for each of the deployment cases could then be extracted from the available data. To set the number of factors, the proposed iterative approach described in Section 3.2.3 was adopted. Figure 6 shows the variations induced by the adopted number of factors on the variance of the decision-making output (i.e., the JS-divergence values), for the four sensor deployment cases. In the charts, the optimal values of the factors to be accounted for in the analysis, constrained by the condition *n_f_* ≤ *n_s_*, can be then clearly identified.

Once the residual matrices were determined, the third level of the proposed machine learning method was initiated by estimating the probability distributions of the residual samples via the KDE. Figure 7 reports some exemplary results regarding the estimated probability distributions of the residual samples at Sensor 4 location, to compare the distributions relevant to states HC1 and HC3, and states HC1 and DC1 in the second deployment scenario. Figure 7a shows that the estimated probability distributions related to states HC1 and HC3 are roughly similar, whereas there is a clear difference between the distributions related to states HC1 and DC1. Note that HC3 was considered as one of the current states in the monitoring phase, even if undamaged. Results in Figure 7a testify that the proposed method could appropriately manage the problem of environmental variability. On the other hand, a comparison between Figure 7a and Figure 5a testifies that, although the AR coefficients could not properly track the difference between states HC1 and DC1, the proposed method (namely, the distance calculation via the LSD to provide the distance-based features, and the MCMC-FA to remove the environmental variability) gives rise to residual samples featuring an enhanced capability to detect damage.

Finally, the estimated probability distributions were adopted to compute the JS-divergence for early damage detection. First, the distance calculation was used for the normal conditions to provide the distance values and estimate the alarming threshold. Second, the probability distributions relevant to the normal and current states were used to compute the distance values for each current state in the monitoring phase. The results regarding the four deployment cases are shown in Figure 8, where the first 1000 distance values are related to the normal conditions HC1–HC2, and the remaining ones pertain to the states HC3 and DC1–DC5. As can be seen, the first 3000 distance values regarding the undamaged states HC1–HC3 all fall below the threshold limit, accurately labeling these cases as undamaged or normal conditions. Even if HC3 was considered as one of the current states in the monitoring phase, all its distance values result similar to those linked to states HC1 and HC2, below the alarming threshold. The distance values relevant to states DC1–DC5 instead exceed the threshold, highlighting that they represent damaged states of the bridge. For all deployment cases, values of distances increase from state DC1 to state DC5, somehow proportionally to damage severity. Moreover, it is reported that the MCMC and HMC sampler allowed results to be obtained independent of the number of sensors, and with no ambiguity about the possible estimated damage severity.

### 4.3. Comparative Studies

Despite the accurate results achieved in terms of damage detection, it appears necessary to compare the proposed method with some state-of-the-art techniques. The first comparison is related to the use of the classical factor analysis for feature normalization, in place of the proposed MCMC-FA method. For this comparison, the covariance matrices of the training sets **X**_1_–**X**_4_ were estimated, to extract the residual matrices **E_x_** and **E_z_** for each deployment case. These matrices were used to estimate their probability distributions and determine the JS-divergence values. The proposed iterative approach and the extreme value-based technique based on peak-over-threshold were also used to determine the number of factors and estimate the alarming threshold. The results linked to the use of the classical factor analysis are shown in Figure 9. All distance values related to states DC1–DC4 are shown to be above the threshold limit; however, the distance values for state HC3 also exceed this threshold, particularly in Cases 3 and 4. These erroneous results are marked with a red circle and the term *Error I* in the charts. Notice that, in the machine learning literature this kind of error is also defined as false positive or false alarm (namely, distance values regarding the normal condition above the threshold). On this basis, all the distance values for state HC3 above the threshold limits have to be considered false positive errors in decision-making. It is shown that Case 1 in Figure 9a yields a smaller number of false positives, as compared to the other cases. This outcome confirms that the performance of the multi-level algorithm, in conjunction with the classical factor analysis for data normalization, is degraded in the case of a limited number of sensors. Moreover, it is worth remarking that, since all distance values relevant to the damaged cases are above the thresholds, no false negatives (namely, distance values related to damaged states below the threshold) are provided. Further details regarding the performance evaluation of machine learning techniques via false positive and false negative errors, can be found in e.g., [38,61].

Another issue is here related to an inability to remove the effects of environmental variability. As mentioned earlier, the features of each current state were obtained from the information related to states HC1 and HC2. Therefore, the output of each of the current states consists of two parts of 20 samples. In Figure 9, it can be observed that there are inter-state variations too: this error is marked with a blue circle and the term *Error II*. Such latter error implies that the effects of the environmental variability were appropriately handled by the classical factor analysis, most likely due to an improper estimation of the covariance matrices for the training stage caused by the low-dimensional samples. Even if Error II is highlighted only for state DC5, the same can also be observed for the others. In summary, the comparison of the results in terms of damage detection, as provided by the proposed machine learning method resting on the MCMC-FA and by the classical factor analysis, has proven that the former is superior to the latter and provides more reliable results.

Another comparative study was conducted by using the PSD of each vibration signal, as an alternate damage-sensitive feature in place of the AR spectrum. In practical terms, this comparative analysis aims at assessing the performances of parametric and non-parametric spectral-based feature extraction techniques. The proposed multi-level method was then employed again to detect damage, and the corresponding results are shown in Figure 10. Unlike the results of damage detection via the AR spectrum (see Figure 8), it is observed that the PSD provides a low damage detectability along with false positive and false negative errors. Although the MCMC-FA method allows for coping with environmental effects, as testified by the limited discrepancies in the figure within each set of distance quantities relevant to states DC1–DC5, damage detection does not prove acceptable. This outcome demonstrates the superiority of the parametric feature extraction method over the non-parametric one.

Finally, the performance of the proposed method was compared with the conventional machine learning technique based on the Mahalanobis distance and the direct use of the AR coefficients. For this purpose, the AR coefficients of all selected sensors regarding the four deployment cases were collected to build the training and test matrices. The average model orders of states HC1 and HC2 were used, and the Burg method in the time domain was adopted to estimate the model coefficients. Figure 11 shows the relevant results, for which each structural state (either healthy or damaged) is characterized by 20 distance values. Figure 11a shows that the current states DC1–DC5 are accurately detected as damaged conditions, while state HC3 is labeled as undamaged. This means that the classical technique succeeds in detecting damage in the case of all measurements allowed for, namely in the case of densely deployed sensor networks. Conversely, Figure 11b–d proves that the Mahalanobis distance technique with the AR coefficients fails in accurately detecting damage in the case of limited sensor locations, due to numerous false positive and false negative errors. This conclusion then confirms the superiority of the proposed multi-level machine learning technique based on the AR spectrum, for damage detection in the case of a limited number of sensors placed over the structure to be monitored.

## 5. Conclusions

This paper aimed at dealing with challenges related to damage detection in the case of partial structural observations due to a limited deployment of the sensor network, under environmental variability. A parametric spectral-based feature extraction approach based on AR modeling was proposed to estimate the AR spectrum, which was proven to be a reliable damage-sensitive feature even when only limited information on the structural state was exploited. An innovative multi-level machine learning method was then proposed to set the training and test data by the LSD (Level 1), to remove the potential environmental and/or operational variations by an improved factor analysis termed MCMC-FA (Level 2), and to detect early damage by the JS-divergence with the aid of the KDE (Level 3). Experimental datasets relevant to the Wooden Bridge benchmark were exploited to assess the accuracy and performance of the proposed method, and also in comparison with certain alternative exiting approaches.

The results of the present analysis reveal that the AR spectrum, in cooperation with the multi-level machine learning method, can bear a noteworthy potential use to detect early damage in the case of limited sensor deployments and in the presence of environmental effects. This approach was shown to outperform others based on the conventional Mahalanobis distance and direct use of the AR coefficients. The AR spectrum proved to be more effective and reliable than the PSD. It was observed that the proposed MCMC-FA method enhances the performance of the proposed multi-level machine learning, efficiently removing the effects of environmental variability. Moreover, this method was superior to the classical factor analysis, when the size of the multivariate training dataset proves to be small.

In future research activities, the proposed approach is to be extended to long-term monitoring cases still featuring limited sensor locations and environmental variability, with full-scale civil structures such as bridges subjected to ambient vibrations. The impact of the adoption of limited wireless sensor nodes on the damage detection capability will also be thoroughly evaluated. As unsupervised data-driven methods can be employed to locate damage upon conditions of a dense sensor network with an optimal placement of sensors, it appears necessary to investigate this problem in the case of limited sensor coverage. Moreover, it is recommended to investigate the performance of the proposed method using other types of vibration-induced signals, such as e.g., strains and acoustic emission.

## Figures and Tables

**Figure 1 sensors-22-01400-f001:**
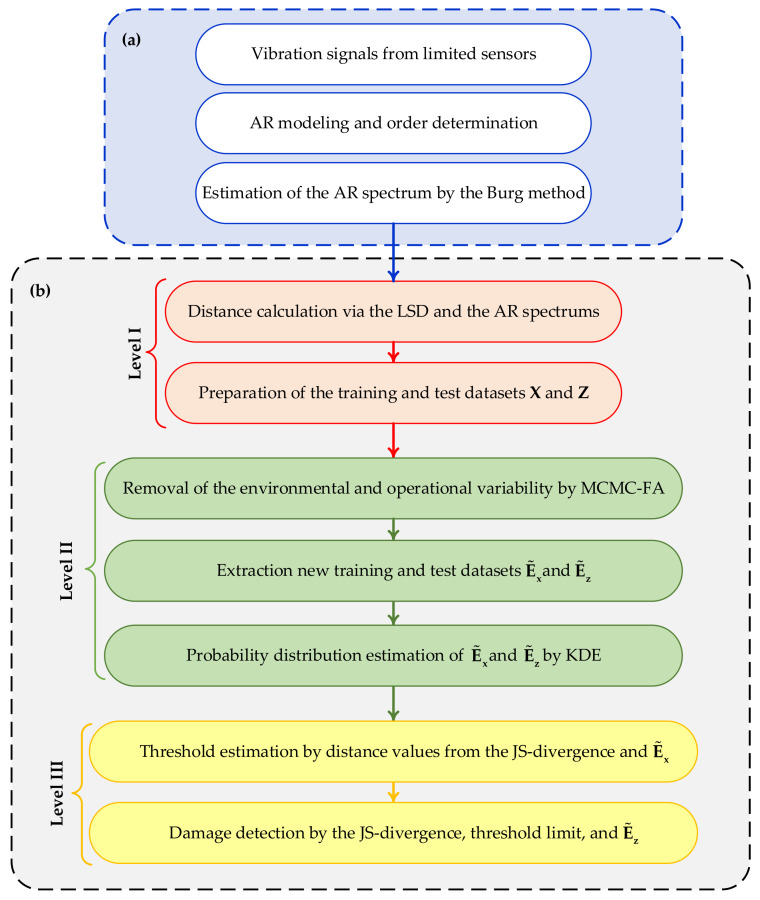
Flowchart of the proposed method: (**a**) feature extraction, (**b**) and multi-level machine learning method.

**Figure 2 sensors-22-01400-f002:**
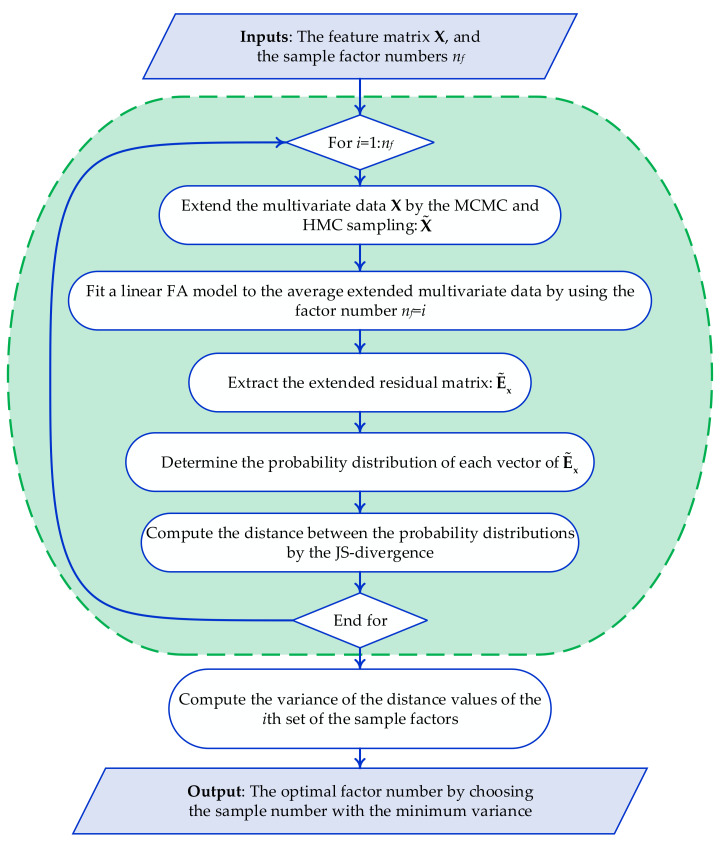
Flowchart of the proposed approach to set the optimal number of factors.

**Figure 3 sensors-22-01400-f003:**
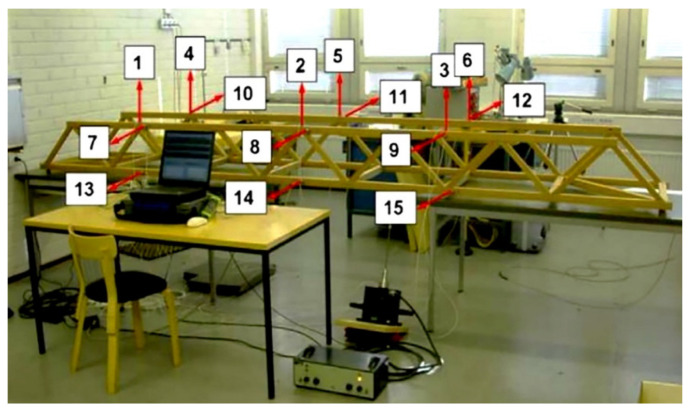
The Wooden Bridge [60].

**Figure 4 sensors-22-01400-f004:**
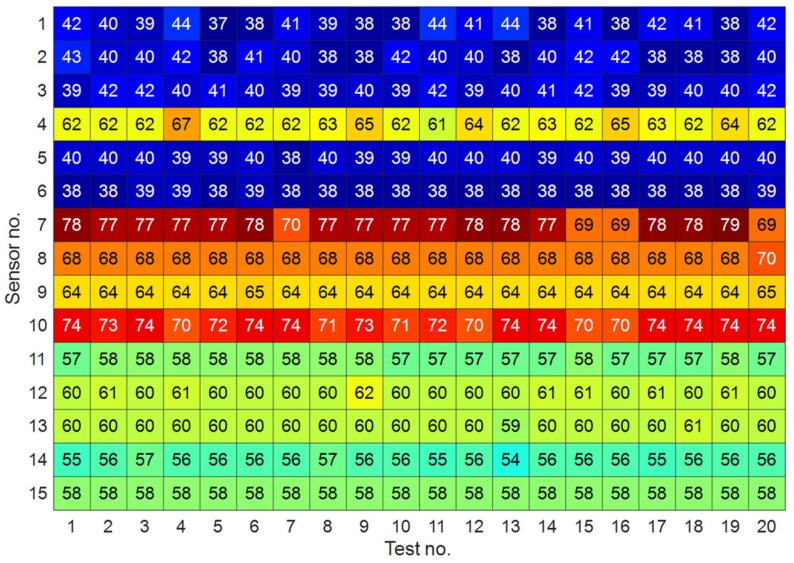
Average AR orders for all sensor locations and test measurements relevant to HC1 and HC2.

**Figure 5 sensors-22-01400-f005:**
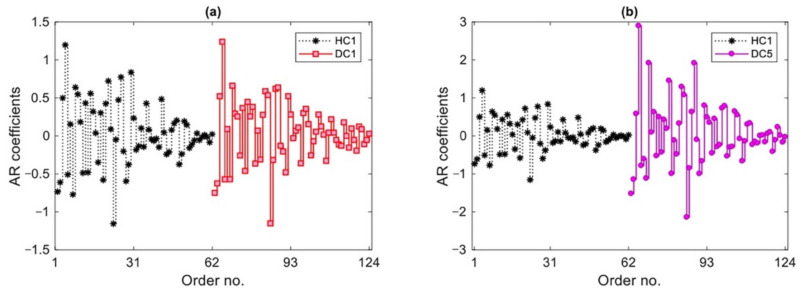
AR coefficients relevant to Sensor 4 for the first test measurement of HC1, DC1 and DC5: (**a**) states HC1 and DC1; (**b**) states HC1 and DC5.

**Figure 6 sensors-22-01400-f006:**
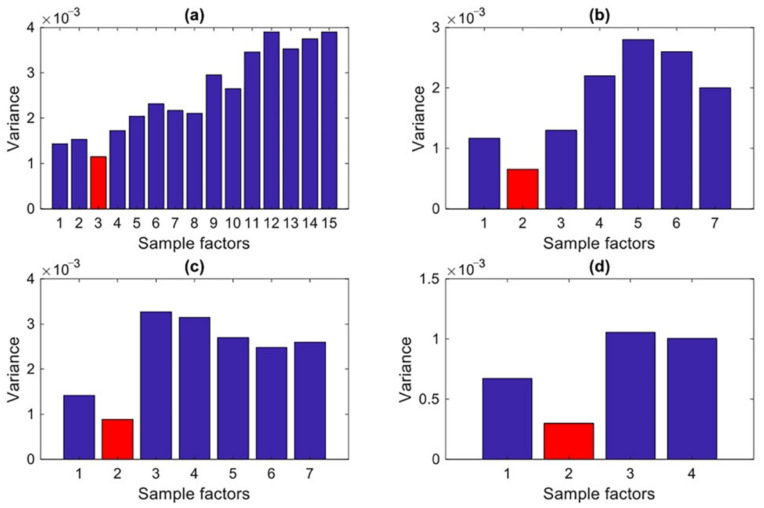
Determination of the number of factors needed for the SFA algorithm: (**a**) Case 1, (**b**) Case 2, (**c**) Case 3, (**d**) Case 4.

**Figure 7 sensors-22-01400-f007:**
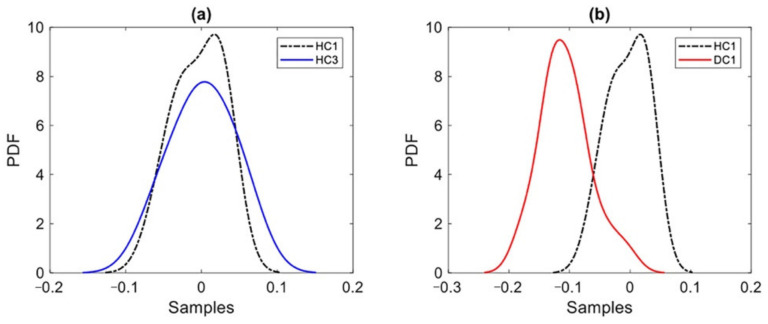
Comparison of the estimated probability distributions of the residual samples relevant to Sensor 4 and to the second deployment scenario: (**a**) states HC1 and HC3; (**b**) states HC1 and DC1.

**Figure 8 sensors-22-01400-f008:**
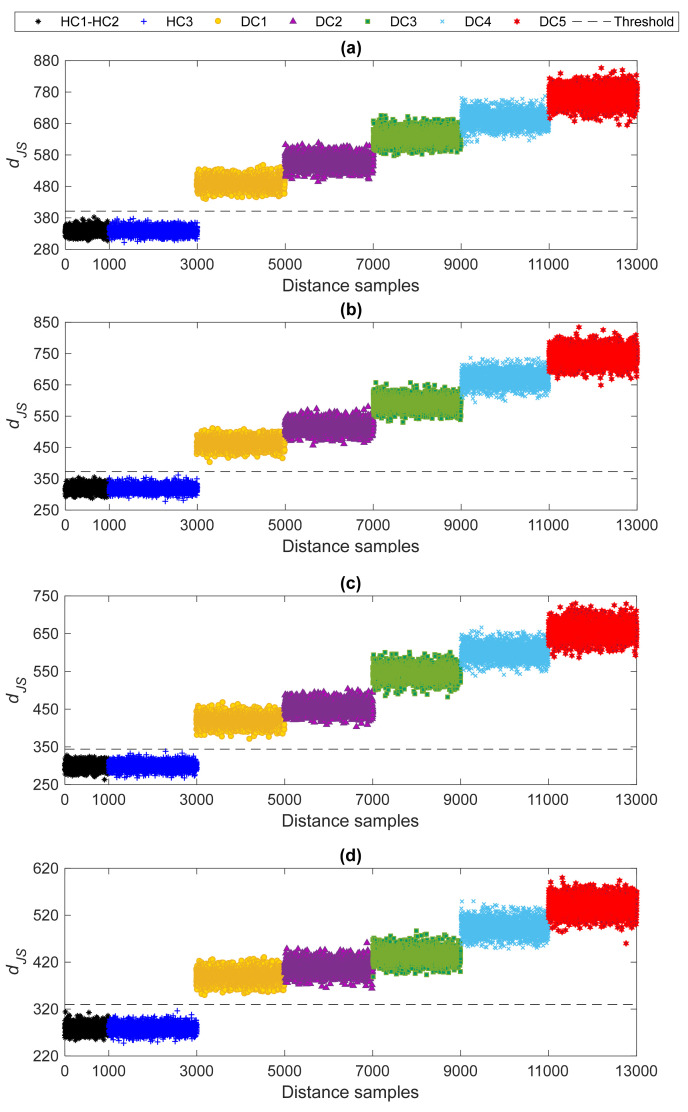
Damage detection by the proposed multi-level machine learning method and AR spectrum: (**a**) Case 1, (**b**) Case 2, (**c**) Case 3, (**d**) Case 4.

**Figure 9 sensors-22-01400-f009:**
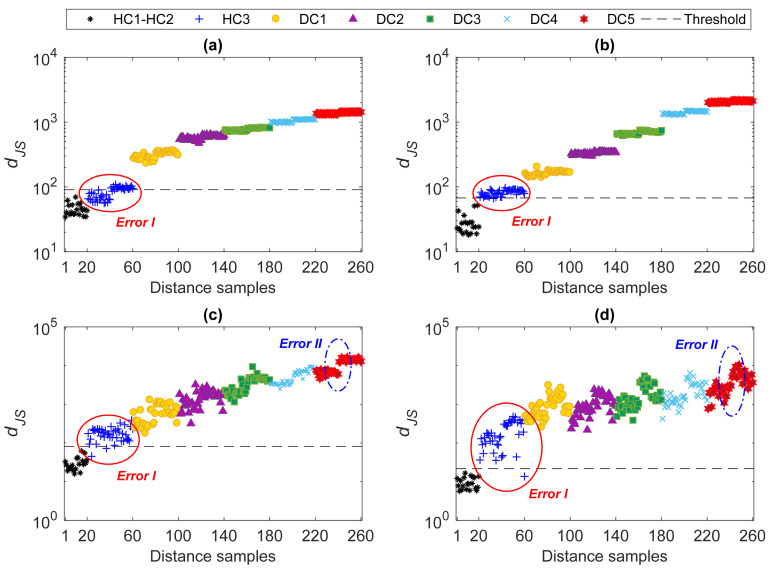
Damage detection by the AR spectrum, classical factor analysis, and JS-divergence: (**a**) Case 1, (**b**) Case 2, (**c**) Case 3, (**d**) Case 4.

**Figure 10 sensors-22-01400-f010:**
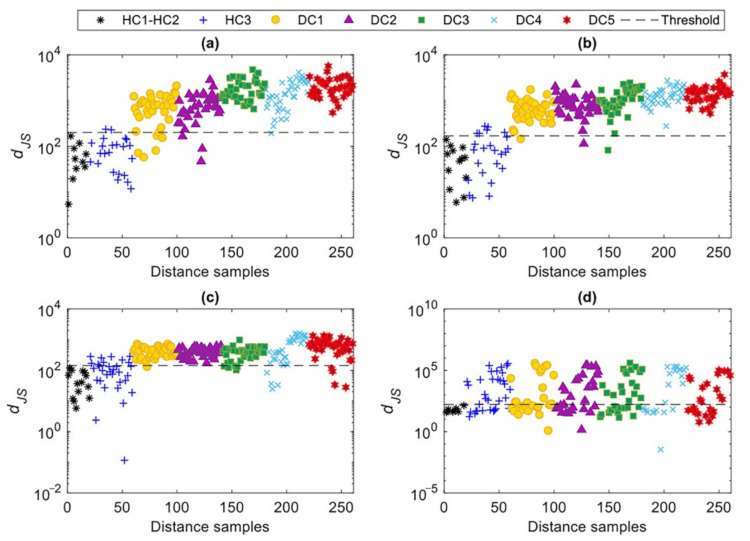
Damage detection by the proposed multi-level machine learning method and PSD: (**a**) Case 1, (**b**) Case 2, (**c**) Case 3, (**d**) Case 4.

**Figure 11 sensors-22-01400-f011:**
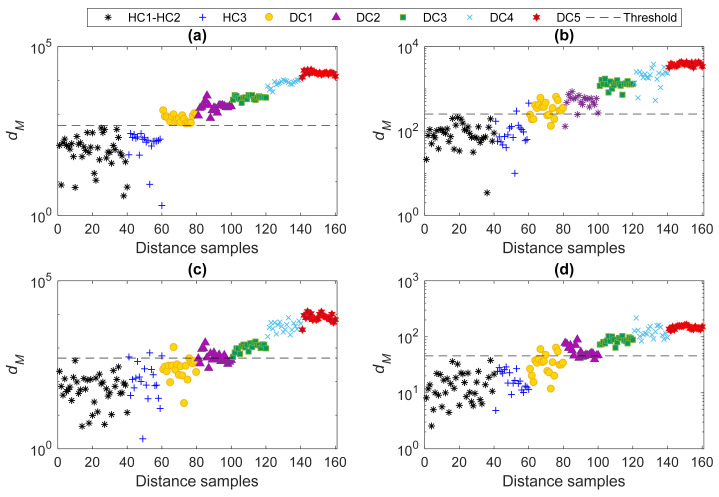
Damage detection by the conventional MSD technique and the AR coefficients: (**a**) Case 1, (**b**) Case 2, (**c**) Case 3, (**d**) Case 4.

**Table 1 sensors-22-01400-t001:** Structural states of the Wooden Bridge.

Day	Condition	Label	Added Mass (g)	Phase
18 May	Undamaged	HC1	-	Baseline
25 May	Undamaged	HC2	-	
29 May	Undamaged	HC3	-	Monitoring
29 May	Damaged	DC1	23.5	
		DC2	47.0	
		DC3	70.5	
		DC4	123.2	
		DC5	193.7	

**Table 2 sensors-22-01400-t002:** Considered sensor deployment scenarios.

Deployment Case	Labels of Active Sensors	Description
1	1–15	100% of deployed sensors
2	2,4,6,7,9,10,14	~50% of deployed sensors
3	1,3,5,8,11,12,15	~50% of deployed sensors with no sensors installed on the damaged area
4	2,5,11,15	~25% of deployed sensors with no sensors installed on the damaged area

**Table 3 sensors-22-01400-t003:** Features of the sampling process based on the MCMC and HMC sampler.

Number of Chains (C)	Number of Samples (N)	Burn-in Value	Probability Type
10	1000	1000	Multivariate Gaussian

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
