# Peer review of "Damage Detection in Largely Unobserved Structures under Varying Environmental Conditions: An AutoRegressive Spectrum and Multi-Level Machine Learning Methodology"

_sensors, 2022, doi:10.3390/s22041400_

Round 1
Reviewer 1 Report
The authors present an innovative SHM methodology based on machine learning. The method builds on the vibration spectrum of a structure to identify damage in a structure. The subject is highly relevant and worth investigating. Also, the research is very elaborate. The text ist well-readable and the English language is of good quality.
In general a nice paper, I suggest to accept the article for publication after a MINOR REVISION. This revision should address at least the following points:
specific comments:
- The very first sentence in the abstract is too long and difficult to grasp. I suggest rephrasing to give the reader an easier access to the paper
- it would be useful to mention already in the abstract that the presented method is a passive system without actuation (alternatively: mention the sources of excitation that are used for the spectrum evaluation).
- the explanation of data-based methods is not sufficiently clear. Please revise.
- line 56: which kind of vibration? Does this refer to the vibration resulting from a damaging event, or is it just structural oscillation that is affected by a damage?
- line 75: indeed the sensitivity to damage or the sensitivity to a damagin event?
- please give some more information about the content of the paper. Which kind of damage shall be detected? Which properties of the damage are measured (location, size, damage type?)
- the authors speak of stage and of levels. To my understanding, both terms refer to the same procedure. I recommend sticking either to the term "stage" or to the term "level" to avoid confusion
- the flowchart in Fig. 2 corresponds to a method in Level II. Therefore I recommend to use green instead of red background color (as in Fig 1).
- the description of the damage detection principle suggests that only the presence of the damage can be detect when the threshold is exceeded. Is it also possible to quantify the damage extent or to determine the damage type? What would be necessary to obtain this information?
- Would it also be possible to apply the method to other kinds of signals (strain, acoustic emission?)
- which kind of damage shall be represented by the additional mass in the case study of the wooden bridge? How is an increased mass representative for a damage which is most likely to reduce the stiffness?
- Which programming environment was used to implement the algorithms? Is it all done Matlab? Please comment whereever appropriate.
Author Response
Reply to Reviewer #1
Dear Reviewer
Thank you very much for your positive opinion and constructive comments. Accordingly, the authors have revised the manuscript. In addressing the comments and suggestions, main changes are highlighted in RED in this new version of the manuscript.
Here below you will find our replies to all the comments.
- The very first sentence in the abstract is too long and difficult to grasp. I suggest rephrasing to give the reader an easier access to the paper.
Reply: Thank you very much for your suggestion. This issue has been corrected.
Lines 12-15:
Vibration-based damage detection in civil structures under data-driven methods requires sufficient vibration responses acquired from various sensors. Due to technical and economic reasons, however, it is not always possible to deploy a large number of sensors over civil structures.
- It would be useful to mention already in the abstract that the presented method is a passive system without actuation (alternatively: mention the sources of excitation that are used for the spectrum evaluation).
Reply: Thank you for your comment. The following expressions have been added to the abstract.
Lines 27-28:
Limited vibration datasets from accelerometers and a shaker excitation source under a passive system relevant to a truss structure have been used to validate the proposed method and compare it with alternate, state-of-the-art strategies
- The explanation of data-based methods is not sufficiently clear. Please revise.
Reply: This issue has been revised.
Lines 52-58:
In contrast to the model-driven techniques that require elaborate finite element models of civil structures, the data-driven methods are only based on raw measured data without any numerical modeling and model updating strategies. In other words, the main objective of the data-driven methods is to discover meaningful information (features) from measured data and then use them for decision-making under the concept of machine learning.
- Line 56: which kind of vibration? Does this refer to the vibration resulting from a damaging event, or is it just structural oscillation that is affected by a damage?
Reply: Any data-based method needs to analyze measured data, extract meaningful information from such data (damage-sensitive features), and develop decision-making systems for damage detection. Hence, one can state that data-based methods work by measured responses acquired from any kind of excitation source, either an event causing damage such as earthquake or structural oscillations caused by wind.
- Line 75: indeed the sensitivity to damage or the sensitivity to a damagin event?
Reply: Thank you for your comment. In general, structural damage directly affects the inherent structural properties of a structure, particularly stiffness. Hence, a damage-sensitive feature also means the information that should be relevant to the structural properties or their variations.
Lines 84-91:
The effectiveness of the SHM system relies on the sensitivity of any extracted feature from the sensed structural responses to damage. This is typically attained with pervasive or dense sensor networks, so that the structural behavior results to be largely observed. It needs to clarify that since structural damage directly affects and changes inherent structural properties, particularly stiffness, a damage-sensitive feature is also relevant to structural properties (stiffness) or their variations.
- Please give some more information about the content of the paper. Which kind of damage shall be detected? Which properties of the damage are measured (location, size, damage type?)
Reply: According to one of the fundamental axioms of SHM, for which a damage detection process needs to compare two structural conditions (Farrar and Worden 2013), the main objective of this manuscript is to detect damage under limited sensor information by comparing undamaged and damaged conditions. Although the main content of this process was fully explained in the introduction, the following information has been added to the revised manuscript.
Lines 120-122:
Hence, the main objective of this research is to evaluate whether the structure with limited sensor information has experienced a damage or it is still in its normal state.
- The authors speak of stage and of levels. To my understanding, both terms refer to the same procedure. I recommend sticking either to the term "stage" or to the term "level" to avoid confusion.
Reply: Thank you very much for your notification. The term “level” has been selected for the proposed method and this issue has been modified in the revised manuscript.
- The flowchart in Fig. 2 corresponds to a method in Level II. Therefore I recommend to use green instead of red background color (as in Fig 1).
Reply: Thank you very much for your attention and kind notification. This issue has been solved.
- The description of the damage detection principle suggests that only the presence of the damage can be detect when the threshold is exceeded. Is it also possible to quantify the damage extent or to determine the damage type? What would be necessary to obtain this information?
Reply: In general, unsupervised data-driven methods can be applied to early damage detection and damage localization, as Farrar and Worden (2013) described in Chapter 13 of their book. For damage quantification and damage type recognition, supervised learning methods as well as model-driven techniques are necessary to apply. As we have proposed a multi-level hybrid learning under unsupervised learning, the main focus has been on early damage detection.
- Would it also be possible to apply the method to other kinds of signals (strain, acoustic emission?)
Reply: Thank you very much for this comment and question. Because we only had acceleration time series of the structure, the reply to this question requires further investigations dealing with such data. Hence, we have mentioned it as further research at the end of the conclusion.
Lines 808-810:
Furthermore, it is recommended to investigate the performance of the proposed method using the other kinds of vibration signals such as strain, acoustic emission, etc.
- Which kind of damage shall be represented by the additional mass in the case study of the wooden bridge? How is an increased mass representative for a damage which is most likely to reduce the stiffness?
Reply: Thank you for your questions. According to one of the fundamental axioms of SHM described by Farrar and Worden (2013), any damage detection requires a comparison
between two structural conditions (Axiom II). For this reason, we have applied the vibration datasets of the wooden bridge to validate our method. In general, structural damage affects the stiffness of the structure, however, in Kullaa (2011), the author simulated structural variations (i.e., undamaged and damaged states) by adding the additional mass. This scenario was also used by Papatheou et al. (2010) and Papatheou et al. (2014).
- Which programming environment was used to implement the algorithms? Is it all done Matlab? Please comment whereever appropriate.
Reply: We applied the MATALB environment for all algorithms.
Lines 512-513:
Note that all algorithms in this study were implemented on the MATLAB R2017a environment.
References
Farrar, C. R., and Worden, K. (2013). Structural Health Monitoring: A Machine Learning Perspective, John Wiley & Sons Ltd, Chichester, West Sussex, United Kingdom.
Kullaa, J. (2011). "Distinguishing between sensor fault, structural damage, and environmental or operational effects in structural health monitoring." Mech. Syst. Sig. Process., 25(8), 2976-2989.
Papatheou, E., Manson, G., Barthorpe, R. J., and Worden, K. (2010). "The use of pseudo-faults for novelty detection in SHM." J. Sound Vib., 329(12), 2349-2366.
Papatheou, E., Manson, G., Barthorpe, R. J., and Worden, K. (2014). "The use of pseudo-faults for damage location in SHM: An experimental investigation on a Piper Tomahawk aircraft wing." J. Sound Vib., 333(3), 971-990.

Reviewer 2 Report
This is a very good paper proposing a multilevel machine learning algorithm for damage detection in structures using limited/partial sensory data. Validation in a laboratory example adds a lot of value to the work. The paper is well written and technically sound. It is therefore recommended for publication. In preparing the final version of the manuscript the authors are invited to consider the following points.
Fiure 1 is barely readable. Same for Fig. 2. Please enlarge text.
Please discuss a little bit more the challenges in the application of this method in the field. What happens with random (e.g. traffic) excitation? How relevant may environmental and operational effects be? How to takle higher level damage identification tasks, such as localization and quantification?
May the authors say something about false positives and false negatives in their classification? How are ROC and precision vs recall curves affected by the limited observation?
A long term application of the proposed method in a real bridge would really represent a major step ahead research.
Author Response
Reply to Reviewer
Dear Reviewer
Thank you very much for your positive opinion and constructive comments. Accordingly, the authors have revised the manuscript. In addressing the comments and suggestions, main changes are highlighted in RED in this new version of the manuscript.
Here below you will find our replies to all the comments.
- 1 is barely readable. Same for Fig. 2. Please enlarge text.
Reply: Thank you very much for your comment. Both figures have been re-drawn.
- Please discuss a little bit more the challenges in the application of this method in the field. What happens with random (e.g. traffic) excitation? How relevant may environmental and operational effects be? How to takle higher level damage identification tasks, such as localization and quantification?
Reply: The authors appreciate you for such insightful comments and questions.
In response to your first question, it needs to clarify here that the excitation load applied to the wooden bridge was random vibration produced by a shaker as shown in Figure 3. Regarding the problem of excitation in SHM, there are two general issues that are forced vibrations and ambient vibrations. In time series analysis, it is well-known that time series models without error polynomial function in their equations (i.e., AR, ARX) are suitable for forced-vibration conditions. On the contrary, time series representations with error polynomial functions (i.e., ARMA and ARMAX) are more appropriate for ambient vibration (Carden and Brownjohn 2008; Entezami et al. 2020; Entezami and Shariatmadar 2019). As the excitation load applied to the wooden bridge was based on the forced vibration, we have applied the AR representation for response modeling.
Lines 543-550:
It should be clarified that since the type of excitation source applied to the Wooden Bridge was forced vibration by the shaker (i.e., the measurable excitation condition), the AR representation is applied to model the vibration responses. In the case of ambient vibration (i.e., the unmeasurable excitation condition), one should apply time series models that allocate error polynomial functions such as ARMA and ARARX for response modeling
Regarding your second question, that was fully discussed in Section 3.2. In the problem of SHM, data or feature normalization is the solution to remove the effects of the environmental and operational variations. In this article, we have proposed a hybrid method as a combination of MCMC and factor analysis for this issue (Section 3.2).
Concerning your last question, it should be clarified that unsupervised data-driven methods, such as the method proposed in this manuscript, cannot be applied to quantify the severity of damage, as Farrar and Worden (2013)described in Chapter 13 of their book. For this problem, supervised learning methods as well as model-driven techniques are necessary to apply. In relation to the problem of damage localization using a data-driven method, this process strongly depends on the number of sensors and their deployment. Hence, in the case of using limited sensors, it is necessary to survey this issue. We have mentioned it at the end of the conclusion.
Lines 804-808:
As unsupervised data-driven methods can be applied to locate damaged areas on conditions of a dense sensor network with an optimum sensor placement for damage localization, it is necessary to survey on this problem with limited sensors.
- May the authors say something about false positives and false negatives in their classification? How are ROC and precision vs recall curves affected by the limited observation?
Reply: Thank you very much for your questions. Fortunately, the results of damage detection by the proposed multi-level method provide reliable outputs for decision-making without any false positive and false negative errors. However, the results in Section 4.3 made some errors. For this reason, we have briefly discussed the error rates in the text of the manuscript.
Lines 695-709:
Notice that, in most of the machine learning literature, this kind of error is also defined as false positive or false alarm (namely, distance values regarding the normal condition above the threshold). On this basis, all distance values of HC3 above the threshold limits are false positive errors occurred in decision-making. It is seen that Case 1 in Fig. 9(a) yields fewer false positives compared to the other cases. This conclusion confirms that the performance of the multi-level algorithm in conjunction with the classical factor analysis for data normalization is degraded in case of reducing the number of sensors. It is worth remarking that since all distance values of the damaged cases are over the thresholds, no false negatives (namely, distance values related to damaged states below the threshold) are available. More details about performance evaluations of machine learning techniques via false positive and false negative errors can be found in [33,55].
- A long term application of the proposed method in a real bridge would really represent a major step ahead research.
Reply: The authors appreciate you for this recommendation. It has been mentioned at the end of the conclusion.
Lines 799-802:
In future research activities, the proposed approach is going to be extended to long-term monitoring cases still featuring limited sensor locations and environmental variability, with full-scale civil structures such as bridges subjected to ambient vibrations.
References
Carden, E. P., and Brownjohn, J. M. (2008). "ARMA modelled time-series classification for Structural Health Monitoring of civil infrastructure." Mech. Syst. Sig. Process., 22(2), 295-314.
Entezami, A., Sarmadi, H., Behkamal, B., and Mariani, S. (2020). "Big data analytics and structural health monitoring: A statistical pattern recognition-based approach." Sensors, 20(8), 2328.
Entezami, A., and Shariatmadar, H. (2019). "Damage localization under ambient excitations and non-stationary vibration signals by a new hybrid algorithm for feature extraction and multivariate distance correlation methods." Struct. Health Monit., 18(2), 347-375.
Farrar, C. R., and Worden, K. (2013). Structural Health Monitoring: A Machine Learning Perspective, John Wiley & Sons Ltd, Chichester, West Sussex, United Kingdom.
